# OpenReview forum: "BioDEX: Large-Scale Biomedical Adverse Drug Event Extraction for Real-World Pharmacovigilance"
_EMNLP/2023/Conference — EMNLP 2023 Findings_

### Official Review · Reviewer_4Yca · 2023-08-02

**Soundness:** 3

**Excitement:**

4: Strong: This paper deepens the understanding of some phenomenon or lowers the barriers to an existing research direction.

**Paper Topic And Main Contributions:**

This paper investigates Adverse Drug Event (ADE) extraction, with a focus on drug safety reports. The main contribution is a large dataset of medical papers paired with their corresponding drug safety reports (called BioDEX), along with a descriptive analysis of the dataset (evolution over time, top keywords, distribution by country). The authors also propose a sequence-to-sequence approach to extract key information to automatically populate the safety reports, which they train and evaluate using a subset of the BioDEX dataset, referred to as the Report-Extraction dataset.

**Questions For The Authors:**

A) How were the core attributes chosen? Were medical experts consulted? It seems to me a patient age group attribute would have been valuable (and not too complex) to extract as well.

B) I expect the performance to vary across attributes (with patient sex probably being the easiest to predict), is that the case?

**Reasons To Accept:**

- The paper tackles an important medical application
- The dataset will be valuable for ADE research
- The code to create the dataset and reproduce the experiments is made available

**Reasons To Reject:**

- I'm not sure the formulation of the Report-Extraction task as a sequence-to-sequence task is the most appropriate. Using a classification approach for attributes like seriousness and patient sex seems a better fit. A combination of classification (for seriousness and patient sex) and se2seq (for the other two attributes) would be interesting.
- The evaluation metric is a bit confusing. It is defined in 5.3 as the average over four attributes, two measured by accuracy and two by precision/recall (F1?). In the results section, it is referred to as F1. Which is it?
- The evaluation is not very detailed. I would have liked to see the performance per attribute and not only overall.

**Reproducibility:**

5: Could easily reproduce the results.

**Reviewer Confidence:**

3: Pretty sure, but there's a chance I missed something. Although I have a good feel for this area in general, I did not carefully check the paper's details, e.g., the math, experimental design, or novelty.

---

> ### Author Rebuttal · Authors · 2023-08-28
>
> Thank you for your insightful comments on our work. Your comments specifically helped us improve the soundness of our work, by further elaborating on the metrics we used and performing additional analysis into the model behavior and dataset statistics (which we will include in the next draft of our manuscript). We’re excited about how these changes will help further adoption of BioDEX as a new resource for the important real-world task of pharmacovigilance. Additionally, your comments about modeling formalisms have prompted us to think more deeply about this problem and will influence how we further push performance on this task.
>
> >I'm not sure the formulation of the Report-Extraction task as a sequence-to-sequence task is the most appropriate. Using a classification approach for attributes like seriousness and patient sex seems a better fit. A combination of classification (for seriousness and patient sex) and se2seq (for the other two attributes) would be interesting.
>
> As noted in the review, a (hybrid) classification approach could definitely be explored as an alternative to the sequence-to-sequence formulation we adopted. In this work, we decided to prioritize the sequence-to-sequence approach for the following two reasons:
>
> - We wanted to provide a flexible task setup that could easily be altered by other researchers to incorporate additional attributes. The sequence-to-sequence approach makes this really easy, since you only need to design a new “stringified” representation of the report on which models are trained and evaluated.
>
> - Sequence-to-sequence has become one of the most dominant modeling paradigms in NLP. Models such as the GPT-family can easily be used in this mode. We wanted to offer a clear comparison between specialized, fine-tuned models and these general-purpose, in-context models. Thus, we opted for a sequence-to-sequence formalism.
>
> We are really excited to see how different modeling paradigms compare against our initial approach here, and we encourage other researchers to explore these on this benchmark. To facilitate follow-up work, we invested a lot of effort in producing reproducible, documented, and sound code (which can be found in the supplementary material).
>
> >The evaluation metric is a bit confusing. It is defined in 5.3 as the average over four attributes, two measured by accuracy and two by precision/recall (F1?). In the results section, it is referred to as F1. Which is it?
>
> For every instance of our dataset, a model predicts two sets (drugs and reactions) and two labels (seriousness and patient sex). We can treat these label predictions each as a singleton-set prediction for a uniform protocol across all of them. On these two label attributes, the accuracy (either 0 for a wrong prediction or 1 for a correct prediction) is the same as the precision / recall if we would treat it as a singleton-set. Thus, the accuracy on these two labels represents both the precision and recall on these attributes on the instance-level. Thus, we can aggregate a precision, recall and F1 score per instance by taking a weighted average over the 4 attributes. Now that we have an F1 score per instance, we further take an unweighted average over all the instance-level F1 scores to calculate a dataset-level macro F1 score. This score is reported in the tables.
>
> We agree that the terminology used in our writing was confusing. Based on your comment, we will more thoroughly explain the exact steps we took to reach our final metric.
>
> > The evaluation is not very detailed. I would have liked to see the performance per attribute and not only overall.
>
> Based on this suggestion, we performed an additional analysis with regard to the 4 core attributes. We found these results to give crucial additional information into the model behavior. Thus, these results will be included in the next version of our draft.
>
> Below, we’ve attached the attribute-level performances for the best Flan-T5 and best GPT run on the validation set.
>
> While FLAN-T5 and GPT perform similarly on the seriousness and patientsex predictions, GPT is severely lacking when predicting drugs and reactions compared to FLAN-T5. We believe that FLAN-T5 has an easier time learning the correct terms used to describe drugs and reactions, together with their prior distribution in the FAERS data. GPT, on the other hand, can only learn the task from a few in-context examples and might struggle in the face of these vast drug and reaction ontologies.
>
> This attribute-level breakdown highlights the benefits of using a custom, fine-tuned model over in-context learning. Additionally, a way to improve the in-context learning performance would be to focus on more efficiently learning these challenging and vast drug and reaction ontologies. We will include this breakdown and corresponding discussion in a new version of the paper.
>
> | |FLAN-T5| | |GPT| | |
> |:----|:----|:----|:----|:----|:----|:----|
> | |accuracy|precision|recall|accuracy|precision|recall|
> |seriousness|92.90|/|/|94.00|/|/|
> |patientsex|92.65|/|/|93.00|/|/|
> |drugs|/|62.75|60.94|/|50.40|51.59|
> |reactions|/|38.31|30.78|/|14.17|11.38|
>
>
>
> > How were the core attributes chosen? Were medical experts consulted? It seems to me a patient age group attribute would have been valuable (and not too complex) to extract as well.
>
> The core attributes were chosen based on the expert-designed FAERS reports, which form the basis of all pharmacovigilance reporting. We consulted pharmacovigilance experts on the meaning of these attributes. However, not all attributes in the FAERS reports are always reported. While the patient age group is clearly an important attribute from the medical perspective, it is present in only 18% of the reports. Contrast this with the patient sex attribute which is present in 75% of reports. For this first round of pharmacovigilance models, we opted to focus on the most generic and widely available attributes. However, because of the size of our resource, follow-up work that investigates less frequent attributes such as the patient age group or patient weight (only filled-in 5% of the reports) will most likely still have enough data to train and evaluate meaningful models.
>
> Motivated by this question, we’ve calculated for each attribute the frequency of occurrence in our dataset and we will add these statistics as supplementary information in the next version of this draft. We hope this allows researchers to more efficiently gauge if the BioDEX resource could be applicable to their use case of interest.
>
> >  I expect the performance to vary across attributes (with patient sex probably being the easiest to predict), is that the case?
>
> This is indeed the case. Patient sex and seriousness are the easiest to predict, while predicting the drugs and reactions is significantly more challenging. See our answer on the attribute-level breakdown for a detailed discussion with accompanying results.

---

### Official Review · Reviewer_aAaA · 2023-08-05

**Typos Grammar Style And Presentation Improvements:** Line 446, should it be 50.44 -> 50.45?
**Soundness:** 3

**Excitement:**

4: Strong: This paper deepens the understanding of some phenomenon or lowers the barriers to an existing research direction.

**Paper Topic And Main Contributions:**

This paper creates a large-scale dataset for Biomedical adverse Drug Event Extraction, which contains various attributes. The authors further investigate the performance of pretrained language models in the few-shot and fine-tuning settings, showing the challenges in extracting structural information from the proposed biomedical dataset and the potential in aiding pharmacovigilance.

**Reasons To Accept:**

1. A large-scale corpus for Biomedical adverse Drug Event Extraction is created and will be beneficial to a broader community.
2. Experiments demonstrate the potential of large language models applying to the biomedical domain, either with a few-shot or finetuning.
3. A further discussion on how large language models would benefit pharmacovigilance.

**Reasons To Reject:**

1. Further analysis of the experiment is needed. The task is to predict 4 attributes and the paper report the overall performance. What is the breakdown performance for each attribute? That would be beneficial in understanding the models.
2. The color of the figures is confusing.
2.1 In Figure 5, what does the color mean for each area/country? Does the color indicate the number of reports? and why only a few have the actual report number? Providing a sidebar would help with the understanding.
2.2 Similarly in Figure 6, what is the color of the dot? Assuming light blue is baseline but it is still not clear enough.
3. For the ethical consideration, age and gender could also be the confounder, which is worth mentioning.

**Reproducibility:**

4: Could mostly reproduce the results, but there may be some variation because of sample variance or minor variations in their interpretation of the protocol or method.

**Reviewer Confidence:**

4: Quite sure. I tried to check the important points carefully. It's unlikely, though conceivable, that I missed something that should affect my ratings.

---

> ### Author Rebuttal · Authors · 2023-08-28
>
> Thank you for your insightful comments on our work. Your comments helped us to perform additional analysis, increase the accessibility of our manuscript, and provide additional guidance regarding responsible use of this resource. We believe these changes will help establish BioDEX as the largest and most naturalistic resource for the important real-world task of pharmacovigilance.
>
> Specifically the question about the model performance breakdown with regard to the 4 core attributes motivated us to perform an additional analysis. These new results ended up giving us valuable insights into the model behavior, thus improving the soundness of our analysis. We will include the new results and the accompanying discussion (see our detailed answer to your question below) in the next version of our draft.
>
> Additionally, the code to run this new analysis will be added to our codebase, which you can find in the supplementary material. The readme-file features detailed instructions to reproduce all of our data and results, and our code is designed to be easily extended by independent researchers to perform these kinds of additional analysis and audit the soundness of our work.
>
> > Further analysis of the experiment is needed. The task is to predict 4 attributes and the paper report the overall performance. What is the breakdown performance for each attribute? That would be beneficial in understanding the models.
>
> Based on this suggestion, we performed an additional analysis with regard to the 4 core attributes. We found these results to give crucial additional information into the model behavior. Thus, these results will be included in the next version of our draft.
>
> Below, we’ve attached the attribute-level performances for the best Flan-T5 and best GPT run on the validation set.
>
> While FLAN-T5 and GPT perform similarly on the seriousness and patientsex predictions, GPT is severely lacking when predicting drugs and reactions compared to FLAN-T5. We believe that FLAN-T5 has an easier time learning the correct terms used to describe drugs and reactions, together with their prior distribution in the FAERS data. GPT, on the other hand, can only learn the task from a few in-context examples and might struggle in the face of these vast drug and reaction ontologies.
>
> This attribute-level breakdown highlights the benefits of using a custom, fine-tuned model over in-context learning. Additionally, a way to improve the in-context learning performance would be to focus on more efficiently learning these challenging and vast drug and reaction ontologies. We will include this breakdown and corresponding discussion in a new version of the paper.
>
> | |FLAN-T5| | |GPT| | |
> |:----|:----|:----|:----|:----|:----|:----|
> | |accuracy|precision|recall|accuracy|precision|recall|
> |seriousness|92.90|/|/|94.00|/|/|
> |patientsex|92.65|/|/|93.00|/|/|
> |drugs|/|62.75|60.94|/|50.40|51.59|
> |reactions|/|38.31|30.78|/|14.17|11.38|
>
>
> > The color of the figures is confusing. 2.1 In Figure 5, what does the color mean for each area/country? Does the color indicate the number of reports? and why only a few have the actual report number? Providing a sidebar would help with the understanding. 2.2 Similarly in Figure 6, what is the color of the dot? Assuming light blue is baseline but it is still not clear enough.
>
> As suggested, we will add a color bar with accompanying label to Figure 5. Additionally, we will expand the caption to explain that we only display the actual report number for a handful of countries, so as to not overly clutter our graph. For Figure 6, we will also expand the caption to highlight that the lighter color is the baseline model. We want to position BioDEX as a cornerstone resource for pharmacovigilance. Hence, we are really grateful for these comments which improve the accessibility of our manuscript.
>
> > For the ethical consideration, age and gender could also be the confounder, which is worth mentioning.
>
> Indeed, the age and gender, as well as other attributes captured in our dataset (weight, disease onset stage, etc.) can act as confounders. Before utilizing our model in real-world applications, it is paramount that the performance of our model with regard to these confounders is well studied. We will add a sentence to the ethical consideration section, reflecting these potential additional confounders as suggested.
>
> > Line 446, should it be 50.44 -> 50.45?
>
> This is correct and we will fix this mistake in our next draft. Thank you for spotting this mistake.

---

### Official Review · Reviewer_Qsyq · 2023-08-05

**Soundness:** 4

**Excitement:**

4: Strong: This paper deepens the understanding of some phenomenon or lowers the barriers to an existing research direction.

**Missing References:**

N/A

**Paper Topic And Main Contributions:**

- This paper introduces BioDEX, a resource for Biomedical adverse drug event extraction. BioDEX comprises of pubmed abstracts as well as full text papers along with document-level drug safety reports.
- The paper evaluates the ability of language models to provide the core information of a drug safety report (such as reported weight, age, drug dosages, etc.) given the full text article.
- Models evaluated on BioDEX were found to have lower F1 score than human medical experts

**Questions For The Authors:**

- In Section 6.1, it is mentioned that long papers were truncated to fit the context window. Would it be possible split these papers into smaller chunks and evaluate each chunk separately?
- Was performance of the language models evaluated on each of the 4 core attributes?
- For the GPT models, was the extent of hallucination measured / quantified?

**Reasons To Accept:**

- The paper addresses a very important challenge for pharmacovigilance reporting and the created BioDEX dataset is a valuable resource in this regard.
- The overall experimental set up is well-defined and the test scenario (i.e. testing on data published chronologically after the data in the training set) mimics real-world applications.
- The paper also demonstrates/discusses how their solutions can be immediately applicable towards augmenting the existing pharmacovigilance reporting process



**Reasons To Reject:**

No reason to reject

**Reproducibility:**

4: Could mostly reproduce the results, but there may be some variation because of sample variance or minor variations in their interpretation of the protocol or method.

**Reviewer Confidence:**

4: Quite sure. I tried to check the important points carefully. It's unlikely, though conceivable, that I missed something that should affect my ratings.

**Typos Grammar Style And Presentation Improvements:**

- In Figure 5, it would be helpful to include a color bar

---

> ### Author Rebuttal · Authors · 2023-08-28
>
> Thank you for your insightful comments on our work. Your comments helped us think about alternative modeling approaches, perform additional analysis, and increase the accessibility of our manuscript. We believe these changes will help us make the case for BioDEX as a cornerstone resource for the important real-world task of pharmacovigilance.
>
> The first three paragraphs below discuss your questions and the impact they’ve had on our work at a high-level. The subsequent paragraphs contain detailed in-line responses to your questions.
>
> The questions about context windows and model hallucinations have prompted us to think more deeply about alternative modeling approaches (we will provide detailed answers to your specific questions below). We are excited to continue pushing performance on this important yet challenging real-world task forward, specifically taking these constraints of context windows and hallucinations into account as noted by you.
>
> The question about the model performance breakdown with regard to the 4 core attributes motivated us to perform an additional analysis. These new results ended up giving us valuable insights into the model behavior, thus improving the soundness of our analysis. We will include the new results and the accompanying discussion (see our detailed answer to your question below) in the next version of our draft.
>
> Additionally, the code to run this new analysis will be added to our codebase, which you can find in the supplementary material. The readme-file features detailed instructions to reproduce all of our data and results, and our code is designed to be easily extended by independent researchers to perform these kinds of additional analysis.
>
> > In Section 6.1, it is mentioned that long papers were truncated to fit the context window. Would it be possible to split these papers into smaller chunks and evaluate each chunk separately?
>
> Yes. There are many potential ways of mitigating the fixed context window. As noted, evaluating separate chunks and aggregating predictions is one possible way of incorporating more context. Retrieval-augmented models could also help process these long papers more efficiently. We are excited to see what methods perform best on BioDEX, and how much they improve upon our baseline of truncating the papers to fit in the context window. We invested a lot of effort in writing clean, well-documented, and reproducible code (which you can find in the supplementary material) which allows for easy implementation and benchmarking of these alternative modeling approaches.
>
> > Was performance of the language models evaluated on each of the 4 core attributes?
>
> Based on this question, we performed an additional analysis with regard to the 4 core attributes. We found these results to give crucial additional information into the model behavior. Thus, these results will be included in the next version of our draft.
>
> Below, we’ve attached a table containing the attribute-level performances for the best Flan-T5 and best GPT run on the validation set.
>
> While FLAN-T5 and GPT perform similarly on the seriousness and patientsex predictions, GPT is severely lacking when predicting drugs and reactions compared to FLAN-T5. We believe that FLAN-T5 has an easier time learning the correct terms used to describe drugs and reactions, together with their prior distribution in the FAERS data. GPT, on the other hand, can only learn the task from a few in-context examples and might struggle in the face of these vast drug and reaction ontologies.
>
> This attribute-level breakdown highlights the benefits of using a custom, fine-tuned model over in-context learning. Additionally, a way to improve the in-context learning performance would be to focus on more efficiently learning these challenging and vast drug and reaction ontologies. We will include this breakdown and corresponding discussion in a new version of the paper.
>
> | |FLAN-T5| | |GPT| | |
> |:----|:----|:----|:----|:----|:----|:----|
> | |accuracy|precision|recall|accuracy|precision|recall|
> |seriousness|92.90|/|/|94.00|/|/|
> |patientsex|92.65|/|/|93.00|/|/|
> |drugs|/|62.75|60.94|/|50.40|51.59|
> |reactions|/|38.31|30.78|/|14.17|11.38|
>
>
> > For the GPT models, was the extent of hallucination measured / quantified?
>
> If the GPT models output wrong predictions, this is reflected in a lower F1 score (in particular a lower precision). Thus, hallucinations (instances where the model generates false drugs or reactions) are reflected in a lower score. One potential way of reducing hallucinations would be to first perform an extractive step where the GPT models can only draw on spans of text from the original publication. That way, all the drug and reaction mentions output are explicitly grounded in the original text. This would be an interesting follow-up study, the success of which can be benchmarked with our introduced metrics and data.
>
> > In Figure 5, it would be helpful to include a color bar
>
> Agreed, this will increase the accessibility of our work. We will include a color bar in the next version of our draft.

---

### Meta-Review · Area_Chair_fBak · 2023-09-15

**Recommendation:** 4

**Metareview:**

This paper describes the creation of a large-scale dataset for adverse Drug Event Extraction, within the biomedical domain, encompassing multiple attributes. The dataset is then used employed in an experimental evaluation of some Language Models (LMs). The results show that extracting structural information from the biomedical dataset is still challenging. The study also shows the potential high impact for aiding pharmacovigilance. Reviewers and authors constructively engaged during the rebuttal/discussion phase leading to an potential enhancement of the work. It is highly recommended that the authors integrate reviewers comments.

**Pros**
- the topic addressed is highly relevant for real-world application in pharmacovigilance reporting, and timely;
- the large dataset produced is valuable for research in this field;
- the methodology adopted is robust and well defined;
- readily applicability in the pharmacovigilance reporting applications.

**Cons**
- the breakdown performance for each attribute is not provided in the paper, but authors seem to be ready to include such details in the revised paper;
- the motivation for framing the issue as a sequence-to-sequence task is weak and should be better motivated.
- the definition of the evaluation metric appears to be unclear

---

### Decision · Program_Chairs · 2023-10-07

**Decision:**

Accept-Findings

**Comment:**

This paper describes the creation of a large-scale dataset for adverse Drug Event Extraction, within the biomedical domain, encompassing multiple attributes. The dataset is then used employed in an experimental evaluation of some Language Models (LMs). The results show that extracting structural information from the biomedical dataset is still challenging. The study also shows the potential high impact for aiding pharmacovigilance. Reviewers and authors constructively engaged during the rebuttal/discussion phase leading to an potential enhancement of the work. It is highly recommended that the authors integrate reviewers comments.

**Pros**
- the topic addressed is highly relevant for real-world application in pharmacovigilance reporting, and timely;
- the large dataset produced is valuable for research in this field;
- the methodology adopted is robust and well defined;
- readily applicability in the pharmacovigilance reporting applications.

**Cons**
- the breakdown performance for each attribute is not provided in the paper, but authors seem to be ready to include such details in the revised paper;
- the motivation for framing the issue as a sequence-to-sequence task is weak and should be better motivated.
- the definition of the evaluation metric appears to be unclear